# Identifying Predictors for the Acquisition of Tolerance to Cow’s Milk Protein in Infants with Food Protein-Induced Allergic Proctocolitis (FPIAP): Multifactorial Analysis of Two Italian Cohorts

**DOI:** 10.3390/nu18010095

**Published:** 2025-12-27

**Authors:** Andrea Scavella, Cristina Ferrigno, Mario Baù, Alessandra Colombo, Claudia Ivonne Tavernelli, Marianna Zobele, Roberta Borgetto, Alessandra Maggi, Alice Baronti, Antonio Francone, Gian Vincenzo Zuccotti, Massimo Agosti, Enza D’Auria, Silvia Salvatore

**Affiliations:** 1Post Graduate School of Pediatrics, University of Insubria, 21100 Varese, Italy; ascavella@studenti.uninsubria.it (A.S.); m.bau1@studenti.uninsubria.it (M.B.); citavernelli@studenti.uninsubria.it (C.I.T.); rborgetto@studenti.uninsubria.it (R.B.); abaronti@studenti.uninsubria.it (A.B.); afrancone@studenti.uninsubria.it (A.F.); 2Department of Pediatrics, Vittore Buzzi Children’s Hospital, 20154 Milan, Italy; cristina.ferrigno@unimi.it (C.F.); alessandra.colombo@unimi.it (A.C.); marianna.zobele@unimi.it (M.Z.); alessandra.maggi@unimi.it (A.M.); gianvincenzo.zuccotti@unimi.it (G.V.Z.); 3Department of Biomedical and Clinical Sciences, University of Milan, 20157 Milan, Italy; 4Department of Medicine and Surgery, University of Insubria, 21100 Varese, Italy; massimo.agosti@uninsubria.it; 5Pediatric Unit, Department of Medicine and Technological Innovation, Hospital “F. Del Ponte”, University of Insubria, 21100 Varese, Italy

**Keywords:** non-IgE-mediated food allergy, food protein-induced allergic proctocolitis, FPIAP, rectal bleeding, elimination diet, cow’s milk, food tolerance, infants

## Abstract

**Background/Objectives**: Food protein-induced allergic proctocolitis (FPIAP) is a non-IgE-mediated gastrointestinal food allergy. Although tolerance to the culprit food is usually achieved within the first year of life, late acquisition occurs and remains poorly predictable. This study aimed to analyze clinical characteristics and explore factors that may potentially function as predictors of late tolerance acquisition to cow’s milk (CM). **Methods**: We conducted a cross-sectional study at two Italian pediatric clinics (2020–2024), including infants diagnosed with FPIAP. Clinical, dietary, and immunological variables; onset and duration of rectal bleeding (visible blood in the stools); and time to CM tolerance were analyzed. Late tolerance was defined as acquisition after 19 months according to the distribution of tolerance achievement in our population. Statistical analyses included χ^2^, Mann–Whitney U, Spearman’s correlation, and logistic regression. **Results:** Ninety-four infants were included (median age at onset 2.9 months [IQR 1.9–4.7]); 58 (62%) were exclusively breastfed and 18 (19%) were born preterm (<37 completed weeks of gestation). CM was the culprit food in all cases; tolerance was achieved in all infants at a median age of 12 months. Family history of atopy and atopic dermatitis were reported in 44% and 19% of infants, respectively. Late CM tolerance was associated with preterm birth, fortification of human milk, early antibiotic exposure, growth faltering, and recurrent infections. Logistic regression identified family history of atopy (OR 5.4 [95% CI 1.2–25.4]; *p* = 0.031), atopic dermatitis (OR 8.2 [1.7–40.7]; *p* = 0.010), rectal bleeding >18 days before elimination diet (OR 5.9 [1.3–27.7]; *p* = 0.023), and IgE sensitization (OR 6.4 [1.2–35.0]; *p* = 0.034) as factors that may potentially function as predictors of late tolerance acquisition to CM. **Conclusions:** Identification of factors that may potentially function as predictors of late tolerance acquisition to CM in infants with FPIAP may help providing a personalized clinical management for these patients.

## 1. Introduction

Food protein-induced allergic proctocolitis (FPIAP) is classified as a non-IgE-mediated gastrointestinal food allergy (non-IgE-GI-FA); it represents one of the most common causes of rectal bleeding in infants [1]. The hallmark symptom is bloody, mucous stools, or hematochezia. The prevalence of FPIAP has been estimated to range from 0.16% to 17% among otherwise healthy infants, with an increasing incidence reported worldwide, and it accounts for up to 64% of cases of bloody stools in infancy [2,3]. Symptoms typically begin within the first three months of life [4,5], with a peak onset around one month of age [6]. T lymphocytes appear to play a central role in the pathogenesis of FPIAP [7]. In addition, pro-inflammatory cytokines such as tumor necrosis factor-alpha (TNF-α) and transforming growth factor-beta (TGF-β) have been implicated in the underlying inflammatory mechanisms. Eosinophils are also thought to contribute significantly to disease pathophysiology, as evidenced by their frequent presence in rectosigmoid biopsy specimens [6,8]. Eosinophils induce Type 2 immunity by releasing cytokines, which in turn may promote eosinophilic inflammation, TH2 differentiation, and ultimately the B-cell class switching to IgE [9]. Although FPIAP is traditionally considered a non-IgE-mediated FA, IgE sensitization has been documented in a subset of affected infants [10]. Moreover, infants with FPIAP have been shown to have approximately twice the odds of developing IgE-mediated cow’s milk allergy during the second year of life [11]. Cow’s milk (CM) is the most frequently implicated trigger food in FPIAP, although egg, soy, and wheat may also be involved in a minority of cases [2]. Disappearance of hematochezia usually occurs within days or weeks from starting the elimination diet, whether achieved through maternal dietary CM exclusion in breast-fed infants (maternal CM-free diet) or extensively protein hydrolyzed formulas and, in severe case, amino-acid-based formulas in formula-fed infants [12]. However, the role and necessity of dietary elimination remain a subject of debate [12]. Given that rectal bleeding is often mild, self-limiting, and may resolve spontaneously, a “watch and see” approach for 2 to 4 weeks has been proposed, particularly in exclusively breastfed infants. Dietary elimination is generally recommended in cases of severe rectal bleeding, anemia, or significant parental concern. Nevertheless, there is currently no consensus regarding the optimal indication or duration of allergen elimination [12,13]. Tolerance to CM is typically achieved within the first year of life, although in some infants it may occur earlier, while in others it can be delayed up to three years of age [12]. Despite its generally benign prognosis, FPIAP may negatively affect health outcomes, as it may account for an increased risk of impaired growth and development of functional gastrointestinal disorders [14,15,16]. Recent studies have explored factors potentially influencing the natural history of non-IgE-GI-FA, including age at onset, infant and maternal diet, IgE sensitization, presenting symptoms, intestinal microbiota, early reintroduction of the culprit food, and timing of complementary feeding. However, results remain heterogeneous and often inconclusive [4,10,17,18,19,20,21]. The primary aim of this study was to evaluate clinical and dietary variables in infants diagnosed with FPIAP. The secondary objective was to explore factors associated with food tolerance that may potentially function as predictors of late tolerance acquisition to CM, with the goal of improving a tailored management of these patients.

## 2. Materials and Methods

### 2.1. Study Design and Patients

This is a cross-sectional study that assessed food tolerance acquisition and the clinical characteristics of infants diagnosed with FPIAP according to EAACI and ESPGHAN criteria [13,22] and referred to one of two Italian Pediatric Gastroenterology and Allergology outpatient clinics (“Filippo Del Ponte” Maternal and Child Hospital in Varese and “Vittore Buzzi” Children’s Hospital in Milan), from January 2020 to December 2024.

Patients with FPIAP were eligible for inclusion if they met the following criteria: (1) age between 0 and 12 months at diagnosis; (2) bloody stools as the main or only presenting symptom, in otherwise healthy infants; and (3) availability of clinical and demographic data at diagnosis and subsequent outcome.

Exclusion criteria were: (1) age over 12 months at diagnosis; (2) fever and/or other signs of infection at diagnosis or neonatal enterocolitis; (3) gastrointestinal surgery or anorectal malformation or anal fissures; and (4) known or suspected coagulopathy or hemorrhagic disease.

All parents of enrolled infants were contacted by telephone between January and May 2025, using standardized questions asked by one dedicated clinician from each center, with the aim of collecting detailed information on food tolerance, eventual ongoing allergic manifestations or comorbid conditions, and the overall health status of the infants.

### 2.2. Variable Specification and Measurement

Rectal bleeding was defined as the presence of visible blood in the stools, ranging from minimal blood streaks mixed with the stool to stools containing a substantial amount of blood [13,18].

Diagnosis of CM allergy (CMA) was confirmed with an open oral food challenge (OFC) after a short diagnostic elimination diet (4–8 weeks), according to international guidelines [13,22].

Reintroduction of the culprit food to assess the acquisition of tolerance was scheduled after 6–12 months of diet and was performed in domestic or hospital setting depending on clinical features and family’s compliance. Ladder “modality” (from baked to heated to fresh cheese) was adopted for CM reintroduction in the majority of infants [22,23]; only in selected cases involving fresh dairy products was it directly administered. Timing of tolerance acquisition was defined when the reintroduction of the culprit food did not cause the relapse of the rectal bleeding.

Late tolerance was established according to the distribution of tolerance achievement in our population. Specifically, we based our cut-off on the age of CM tolerance achieved by the third quartile (75th percentile) of our cohort. Therefore, we defined late tolerance as occurring >19 months.

Neonatal data, growth parameters, clinical and demographic features, physical examination at diagnosis, family history, type and duration of diet, timing of food tolerance acquisition, additional allergic manifestation or comorbid conditions and overall health status were retrieved through patients’ electronic medical records and information provided by parents at recall.

Family history of atopy was considered present when at least one of the following allergic conditions occurred in one or more first-degree relatives: allergic asthma, atopic dermatitis, allergic rhinitis-conjunctivitis and IgE-mediated food allergies [24].

Diagnosis of atopic dermatitis (AD) was performed according to Hanifin and Rajka criteria [25,26].

Prematurity was defined according to the World Health Organization (WHO) as birth occurring before 37 completed weeks of gestation or less than 259 days from the first day of the mother’s last menstrual period. We also considered the following subclassification: extremely preterm (<28 weeks), very preterm (28–32 weeks), moderate preterm (32–34 weeks) and late pre-term birth (34–37 weeks) [27].

IgE sensitization was defined as: (1) a positive skin prick test (SPT) to food allergens, indicated by a wheal diameter ≥3 mm compared with the negative control; (2) a specific IgE level ≥0.35 kU/L (ImmunoCAP system, Thermo Fisher Scientific, Uppsala, Sweden) [28].

According to Homan criteria, growth faltering was defined when at least one of the following conditions was met: (1) weight below the 5th percentile for sex and corrected age; (2) weight for length below the 5th percentile; (3) body mass index for age below the 5th percentile; (4) sustained decrease in growth velocity in which weight for age or weight for length/height falls by two major percentiles [29].

Recurrent infections were defined as the occurrence of two or more severe infectious episodes in one year, or three or more respiratory infections in one year, or the need for antibiotics for two months/year [30].

Infants were considered vaccinated against rotavirus in accordance with the Italian National Vaccine Prevention Plan [31], receiving either RotaTeq^®^ (Merck & Co., Inc., Rahway, NJ, USA.; oral three-dose pentavalent) or Rotarix^®^ (GSK Biologicals, Rixensart, Belgiumoral two-dose monovalent) vaccine. Rotavirus vaccination was included as a study variable, since some parents reported the onset of rectal bleeding within a few days after vaccine administration. Based on these observations, we chose to assess this variable to enable a descriptive evaluation of a potential association, without inferring causality. Fortification of human milk was defined as the deliberate addition of proteins or essential micronutrients—specifically, vitamins and minerals—to breast milk, in accordance with the widely recognized definition of food fortification provided by the WHO [32].

As a preliminary step, clinical management and diagnostic procedures were harmonized across the two pediatric centers to ensure methodological consistency and minimize procedural biases. Overall, clinical and diagnostic practices were aligned and comparable between the two outpatient clinics, with the only difference concerning allergy testing: skin prick tests were more frequently performed in Varese, whereas specific IgE measurement was preferred in Milan.

### 2.3. Statistical Analyses

Statistical analyses were performed using JASP (Version 0.95.1, computer software), JASP Team 2025. Categorical data were presented as number values and percentages while non-normally distributed variables as medians and interquartile ranges (IQR). Categorical variables were compared using the χ^2^ test; for continuous variables that deviated from normal distribution, the Mann–Whitney-U test was employed for comparison. We also adopted Spearman’s rank correlation coefficient (rs) to investigate correlation among quantitative variables. A logistic regression model was used to identify potential predictors for late food tolerance development. The results were expressed as odds ratio (OR) and 95% confidence interval (CI). We also indicated performance metrics of the model (AUC, sensibility, specificity, accuracy and precision). Timing for acquisition of food tolerance was identified as the outcome (dependent) variable. Data were represented graphically using box plots and ROC plots. The difference was statistically significant as *p* < 0.05.

## 3. Results

We included 94 patients who met the study criteria, 61 infants from Varese and 33 from Milan. Parents of all recruited infants provided follow-up data at recall (Figure 1).

### 3.1. Demographic and Clinical Findings

The median age at onset of FPIAP was 2.9 months (IQR 1.9–4.7), with no significant differences according to sex; 57% of the infants were male. Nineteen percent of infants were born preterm, and a similar proportion required antibiotic therapy within the first seven days of life for either prophylactic or therapeutic indications.

Atopic dermatitis (AD) and a family history of atopy were reported in 19% and 44% of cases, respectively. Rectal bleeding was present in all infants at diagnosis. Growth faltering was reported in 15%. Other symptoms were less frequent, including vomiting (6%), excessive crying or colic (5%), and diarrhea (3%).

At the first clinical evaluation, 62% of infants were exclusively breastfed, 18% were formula-fed, and 20% received mixed feeding. During the first weeks of life, fortification of breast milk was reported in 13% of cases. CM was identified as the culprit food in all infants; no cases of FPIAP triggered by other or multiple foods were observed.

All patients underwent allergy testing, including serum-specific IgE measurement and/or skin prick testing. Elevated CM–specific IgE levels were detected in 9 of 94 infants (9.6%), while positive skin prick tests were observed in 5 of 94 infants (5.4%).

Recurrent respiratory infections were reported in 16% of patients (15/94). Approximately half of the cohort (54%) had received rotavirus vaccination, and in a small subset of cases parents reported the onset of hematochezia within 1 to 7 days following vaccination.

All infants underwent a therapeutic CM elimination diet. This consisted of a maternal CM–free diet in breastfed infants (62%) and CM-based extensively hydrolyzed formula (eHF) in 32%, amino acid-based formula (AAF) in 5%, and rice-based hydrolyzed formula in 1% of cases.

The demographic and clinical characteristics of our population are shown in Table 1; the main differences between the two cohorts are summarized in Table 2. Differences in the distribution of several variables were observed between the two cohorts, with the Milan cohort showing a higher proportion of male sex, prematurity and fortification of breast milk in the first weeks of life. In addition, differences were observed in feeding modalities and in the type of elimination diet adopted: the Varese cohort showed a predominance of breastfeeding and maternal CM-free diets, whereas the Milan cohort more frequently received formula or mixed feeding, with elimination diets based on extensive hydrolyzed formula (eHF) or aminoacid formula (AAF).

### 3.2. Rectal Bleeding Duration Before Starting Elimination Diet

In our population, the duration of rectal bleeding before starting elimination diet varied substantially, ranging from 0.5 to 72 days. The median duration was 10 days (IQR 2.75–18). We found a statistically significant correlation between rectal bleeding duration before starting elimination diet and timing of CM tolerance achievement (rs 0.325; *p* = 0.001). In addition, considering the third quartile (75th quartile) of rectal bleeding duration before starting elimination diet as a threshold (>18 days), patients who bled for more than this time limit (*n* = 21) presented more frequently late CM tolerance acquisition (*p* = 0.003) (Figure 2).

### 3.3. Age at Onset of FPIAP and Sex

There was no significant correlation between age at onset of rectal bleeding and timing of tolerance acquisition (rs 0.186; *p* = 0.073). Moreover, sex (male *n* = 54) did not affect CM tolerance time (*p* = 0.951).

### 3.4. Neonatal and Growth Data

Preterm newborns (*n* = 18) presented late CM tolerance acquisition more frequently compared to full-term newborns (*p* = 0.009) (Figure 3a), without a significant difference between extremely–very (*n* = 5) and late–moderate preterms (*n* = 13) (*p* = 0.114). A significant association between late CM tolerance and antibiotic treatment within the first seven days of age (*n* = 18) was reported (*p* = 0.002) (Figure 3b). In our cohorts, infants affected by growth faltering presented late CM tolerance more frequently than patients with normal weight growth (*n* = 14) (*p* = 0.017) (Figure 3c).

### 3.5. Atopy and IgE Sensitization

Late CM tolerance achievement was significantly associated with a family history of atopy (*n* = 41) (*p* < 0.001; Figure 4a) and finding of atopic dermatitis (AD) at the first visit (*n* = 18) (*p* < 0.001; Figure 4b) and at CM reintroduction (*n* = 12) (*p* = 0.029). IgE sensitization was detected in 14/94 infants (15%); among these, 5/14 also had concomitant atopic dermatitis. The presence of IgE sensitization was significantly associated with late CM tolerance development (*p* = 0.002) (Figure 4c).

### 3.6. Infant Feeding

Type of feeding reported at the first visit did not affect the timing of CM tolerance acquisition (*p* = 0.746). Likewise, the timing of tolerance acquisition did not differ significantly across the various CM elimination diet approaches (*p* = 0.684). Conversely, we noted a significant association between late CM tolerance achievement and fortification of breast milk in the first weeks of life (*n* = 12) (*p* = 0.015; Figure 5).

### 3.7. Recurrent Infections

In our cohorts, we reported a significant association between late CM tolerance achievement and recurrent infections (*n* = 15) (*p* = 0.003; Figure 6).

### 3.8. Rotavirus Vaccination

No significant association was observed between rotavirus vaccination (*n* = 51) and late tolerance acquisition (*p* = 0.711).

### 3.9. Food Tolerance

All of our patients achieved CM tolerance. The median age at acquisition of tolerance was 12 months (IQR 9.75–18) with a markedly broad range, extending from 6 to 37 months. CM reintroduction was performed using ladder modality in most cases (84%) (Table 3).

### 3.10. Factors Associated with Food Tolerance That May Potentially Function as Predictors of Tolerance Acquisition to CM

Based on the definition previously described, we identified 19 cases (20%) of late acquisition of CM tolerance (>19 months). In the late CM tolerance subgroup, the median age at onset of FPIAP was 3.5 months (IQR 2–5) with no relevant difference regarding biological sex (58% male; *n* = 11).

At the first visit, 53% of infants (*n* = 10) were exclusively breast fed, 21% (*n* = 4) formula fed and 26% breast and formula fed (*n* = 5). CM elimination diet consisted of: maternal CM free diet in 53% of patients (*n* = 10) and CM-based eHF in 47% (*n* = 9). Regarding the clinical features, in addition to rectal bleeding present in all patients, only 11% of infants (*n* = 2) were reported to have episodes of crying/colic; no other symptoms or signs were detected.

According to logistic regression analysis, family history of atopy (OR 5.4 [1.2–25.4]; *p* = 0.031), concomitant atopic dermatitis (AD) (OR 8.2 [1.7–40.7]; *p* = 0.010), rectal bleeding duration >18 days before starting elimination diet (OR 5.9 [1.3–27.7]; *p* = 0.023) and IgE sensitization (OR 6.4 [1.2–35.0]; *p* = 0.034) were adequate factors associated with food tolerance that may potentially function as predictors of late tolerance acquisition to CM (Table 4).

Performance metrics of logistic regression analysis are shown in Table 5 (AUC = 0.888, accuracy = 0.883, precision = 0.833, sensibility = 0.526 and specificity = 0.973).

The ROC curve is illustrated in Figure 7.

Logistic regression analyses were also performed separately within each cohort; due to the small number of events, the estimates were highly unstable and are reported in Appendix A.

## 4. Discussion

To the best of our knowledge, this is the first study to explore multiple factors associated with food tolerance that may potentially function as predictors of tolerance acquisition to CM, exclusively in FPIAP.

The main finding of this study is that family history of atopy, concomitant atopic dermatitis, rectal bleeding duration >18 days before initiation of the elimination diet, and IgE sensitization to CM emerged as potential predictors of late (>19 months) acquisition of CM tolerance, after adjustment for relevant clinical and demographic covariates. In our cohorts, the median age of CM tolerance acquisition was 12 months (IQR 9.75–18), consistent with findings reported in previous studies [10,19,33]. This data should be interpreted with caution considering that, according to the guidelines, the duration of the therapeutic elimination diet was protracted up to 6 months after diagnosis or until the infant was 12 months old, whichever is reached first [13,22]. Hence, earlier tolerance in selected cases could not be excluded.

Moreover, it is worth noting that many of the confidence intervals in the logistic regression analysis in our population are very wide, reflecting the small sample sizes and limited number of events. While some associations reached statistical significance, these estimates should be interpreted with caution due to the limited precision.

Regarding risk factors for late food tolerance in FPIAP, the available literature is markedly heterogeneous and fragmented, and no uniform threshold for defining late tolerance has been established. In a Turkish single center cross-sectional study, earlier onset age and onset of symptoms during breastfeeding both were associated with early tolerance (<12 months old) [17]. More recently, the larger study by Buyuktiryaki et al., conducted in five centers in the same country, reported that IgE sensitization, allergy to multiple foods, and presence of colic were risk factors for persistent course and late tolerance (>12 month) [10]. According to other authors, inadequate maternal diet, concomitant IgE-related food allergy, feeding with CM-based formula, late complementary feeding, type of culprit food (egg and nuts more than CM), use of amino acid-based formula, and presence of diarrhea were related to late tolerance development [5,19,20,33]. Nacaroglu et al. established that infants with FPIAP who did not develop tolerance presented higher mean platelet volume and plateletcrit compared to others with normal tolerance [34].

In our study, infants with a family history of atopy exhibited a 5.4-fold increased risk of developing late tolerance (>19 months). Similar results have been recently reported in “non-IgE-mediated gastrointestinal food allergy (NIGEFA) study” by Carucci et al. who suggested that a family history of allergy was associated with a lower rate of tolerance acquisition at 24 months [18]. However, other authors have reported no evidence supporting a role for this variable in influencing the timing of tolerance acquisition [10].

Moreover, in our cohorts, the presence of AD was associated with an approximately 8.2-fold increased risk of late tolerance. The high prevalence of AD in non-IgE-GI-FAs has been well confirmed in the literature [18,35].

In 2019, Mayer et al. introduced the concept of “non-IgE-mediated Allergic March”, suggesting that AD, asthma, and allergic rhinitis could co-exist in patients with non-IgE-mediated allergies, with an early peak of non-IgE-GI-FAs (including FPIAP) and AD in infancy and a subsequent development of asthma and allergic rhinitis from the time of school-age. The association between non-IgE and IgE mediated-GI-FAs could occur in an overlap allergic march [36].

However, the prevalence of IgE sensitization in infants with a confirmed diagnosis of FPIAP and its correlation with tolerance achievement remain poorly investigated, with most of data coming from the aforementioned Turkish studies.

The NIGEFA study found AD to be the most common concomitant allergic manifestation in non-IgE-GI-FAs [18]. The pathogenic mechanism underlying these forms of allergy could be represented by the involvement of eosinophils that are also detected in rectal biopsies of infants with FPIAP. Eosinophils are known as amplifiers of type 2 immunity; type 2 innate lymphoid cells (ILC2) and TH2 cells produce IL-5, which recruit eosinophils into inflammatory sites and activate the production of eosinophils [37].

In our cohorts, infants with IgE sensitization had a 6.4-fold higher risk of late tolerance (>19 months), in line with previous reports [10,20].

Regarding the duration of rectal bleeding before starting elimination diet, to our knowledge, no evidence has been reported in the literature on the potential relevance of this variable. We observed a statistically significant association between the duration of rectal bleeding before starting elimination diet and the age at which CM tolerance was achieved. Infants with prolonged rectal bleeding (>18 days) had a 5.9-fold higher risk of late tolerance (>19 months). However, in our cohorts, the duration of rectal bleeding before starting elimination diet was highly variable. This variability is likely attributable to several factors, including the amount of rectal bleeding, the degree of parental concern, and the waiting time for specialist evaluation.

We also observed that prematurity was significantly associated with late CM tolerance, although preterm birth did not emerge as a potential predictor in our analysis. It is noteworthy that the proportion of preterm infants in our cohorts was slightly higher than that reported in previous studies (19% vs. 5–16%) [10,18,21]; this difference may be partly attributable to the presence of a NICU and a large neonatology unit within our hospitals. Importantly, in preterm infants presenting with rectal bleeding, establishing a diagnosis of FPIAP can be challenging, given the complex differential diagnosis that includes idiopathic neonatal transient colitis and necrotizing enterocolitis (NEC) [38].

In our study, antibiotic treatment within the first seven days of age was also associated with late CM tolerance, without becoming a predictive factor. Buyuktiryaki et al. previously reported a similar result, considering the use of antibiotics within the first six months of age [10]. A possible interpretation of this variable focuses on the negative effect of antibiotics on the intestinal microbiome. In 2024, a prospective observational study demonstrated that infants affected by FPIAP mostly presented Enterobacteriaceae clusters (specifically Klebsiella and Shigella), typical of an immature microbiome. The reduction of Bifidobacteria from the microbiome in the first months of life may contribute to the manifestation of FPIAP [4]. Antibiotics could have a negative effect on gut microbiota development and possibly predispose to a more severe form of FPIAP, but further large studies are necessary to confirm this hypothesis. In our cohorts, growth faltering and recurrent infections were associated with late CM tolerance while not, however, representing potential predictors. To the best of our knowledge, no previous studies have examined this association accurately. It is well established that weight faltering may occur in non-IgE-GA-FA- and, much less frequently, in FPIAP [10,17]. Diaferio et al. reported that growth faltering may be an awareness sign for early identification of CM allergy, especially in non IgE-GI-FAs [39]. In addition, evidence from an international survey highlighted a relationship between growth parameters and food allergies in infants [14].

Guidelines recommend a close monitoring of growth in children with both CM IgE and non-IgE allergy to identify inadequate nutritional support [22].

The occurrence of recurrent infections has been scarcely investigated in relation to FPIAP. In a recent study, Akbulut et al. reported that infants diagnosed with FPIAP are at high risk of recurrent infections later in life [40]. In agreement with our results, the infections were predominantly respiratory, particularly bronchitis.

In rare cases, rotavirus vaccination has been related to intestinal intussusception, manifesting with persisting inconsolable crying, vomiting and blood and/or mucus in the stool [41], whilst little is known about rectal bleeding occurring in otherwise well-appearing infants. Approximately half of our population was vaccinated against rotavirus and, despite some parents noting hematochezia within a few days from vaccination, this variable was not associated with late acquisition of CM tolerance.

In a more clinically oriented perspective, our findings suggest that several features may help identifying infants who are more likely to experience a late acquisition of CM tolerance. In particular, a prolonged bleeding duration before initiation of the elimination diet (>18 days) and the presence of properly diagnosed atopic dermatitis—particularly when persistent during follow-up—emerge as factors potentially predictive of late tolerance. Therefore, these elements may assist clinicians in anticipating a late tolerance pattern and should be carefully evaluated through an appropriate and structured follow-up.

Although recurrent infections, particularly those of respiratory origin, and growth faltering do not exhibit a potential predictive role, their consistent association with late acquisition of CM tolerance indicates that they exhibit clinical relevance. In practice, this supports the importance of regularly monitoring growth trajectories, ideally according to standardized healthcare protocols, and documenting recurrent infections methodically, in terms of frequency and site of involvement. This level of clinical vigilance may help clinicians to manage these infants more effectively by providing an adequate perspective on their expected timeline for CM tolerance acquisition.

This study has several limitations. First, it presents a small sample size, a cohort-specific variability, a limited number of events, and wide confidence intervals. These factors should be taken into consideration and guide a cautious interpretation of the results presented. Logistic regression analyses were conducted separately within each cohort to explore cohort-specific associations and assess whether the patterns observed in the combined population were consistent across subgroups. However, due to the small sample size and limited number of events, the resulting estimates were highly unstable, confidence intervals were wide, and *p* values were largely uninterpretable. Overall, these observations underscore the importance of considering cohort-specific characteristics when interpreting the results of association analyses, while the primary analyses of our combined population remain informative within the limits of the available data.

Second, due to its cross-sectional design, it is not possible to assess causal relationships or perform formal predictive modelling; the analysis was intended only to explore characteristics that could potentially act as predictors of late acquisition of CM tolerance. Therefore, well-designed prospective longitudinal studies with larger cohorts are needed to verify whether the factors identified in this study truly exert an effective predictive role in late acquisition of CM tolerance. Other limitations include the fact that the study population originates from a single Italian region, which limits the generalizability of our findings. Furthermore, clinical information at recall was based on parent-reported interviews limiting the reliance of data. However, we adopted several measures to mitigate this issue, including the use of patients’ electronic medical records and the administration of specific, targeted recall questions by trained clinicians.

The strengths of this study include the uniform composition of the cohorts, as all enrolled infants presented CM-related FPIAP, the comprehensive assessment of multiple variables, the enrollment of patients from two different pediatric centers including Allergology and Gastroenterology outpatient clinics, and the excellent compliance at recall. However, a longer follow-up period would provide additional information, particularly regarding allergy evolution. Further studies are required to investigate the natural history of FPIAP and to clarify the role of early IgE sensitization.

## 5. Conclusions

This study showed that several factors—namely, family history of atopy, concomitant atopic dermatitis, IgE sensitization, and rectal bleeding duration >18 days before starting elimination diet—emerged as factors that may potentially function as predictors of late tolerance acquisition to CM in FPIAP. These variables can be readily evaluated at the first visit through an accurate medical history, physical examination and the execution of SPTs. Their early identification may support clinicians in initiating an elimination diet more promptly and considering a longer dietary exclusion period in infants presenting with potential predictors of late CM tolerance acquisition. Further studies with larger numbers of patients and longer follow-up are needed to confirm our findings and design a practical algorithm for the tailored management of FPIAP.

## Figures and Tables

**Figure 1 nutrients-18-00095-f001:**
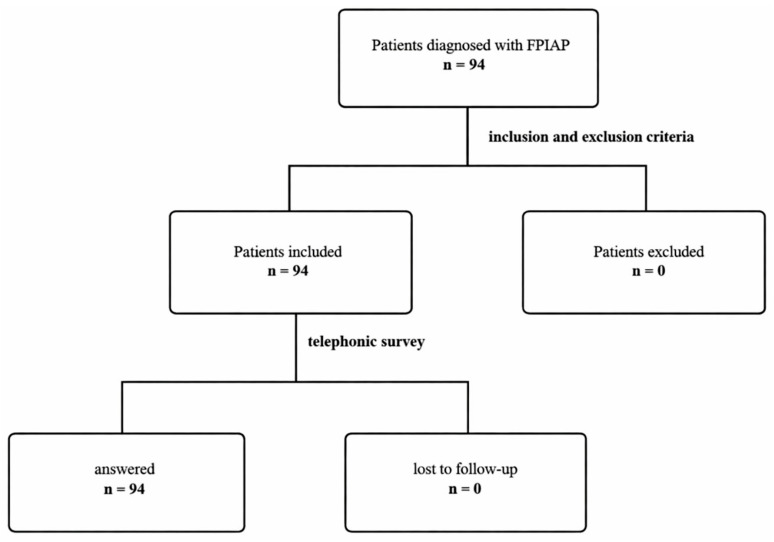
Diagram of participants in the study.

**Figure 2 nutrients-18-00095-f002:**
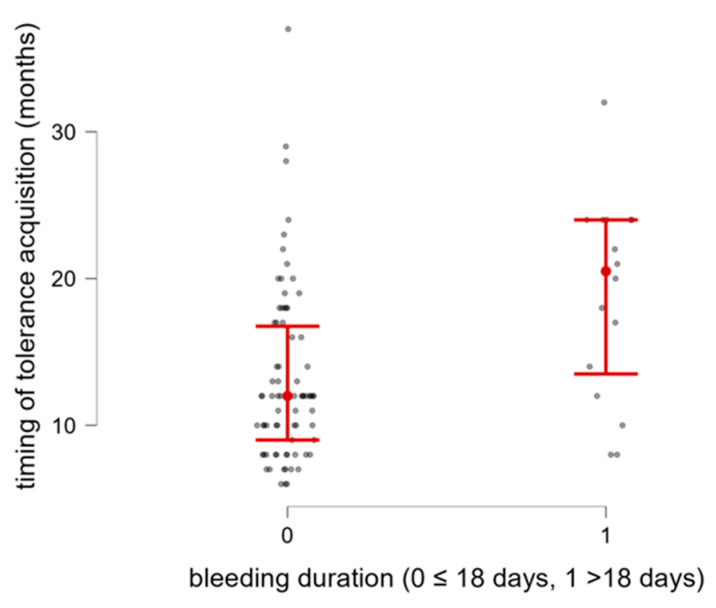
Timing for acquisition of tolerance in relation to rectal bleeding duration before starting elimination diet (0 ≤ 18 days, 1 > 18 days).

**Figure 3 nutrients-18-00095-f003:**
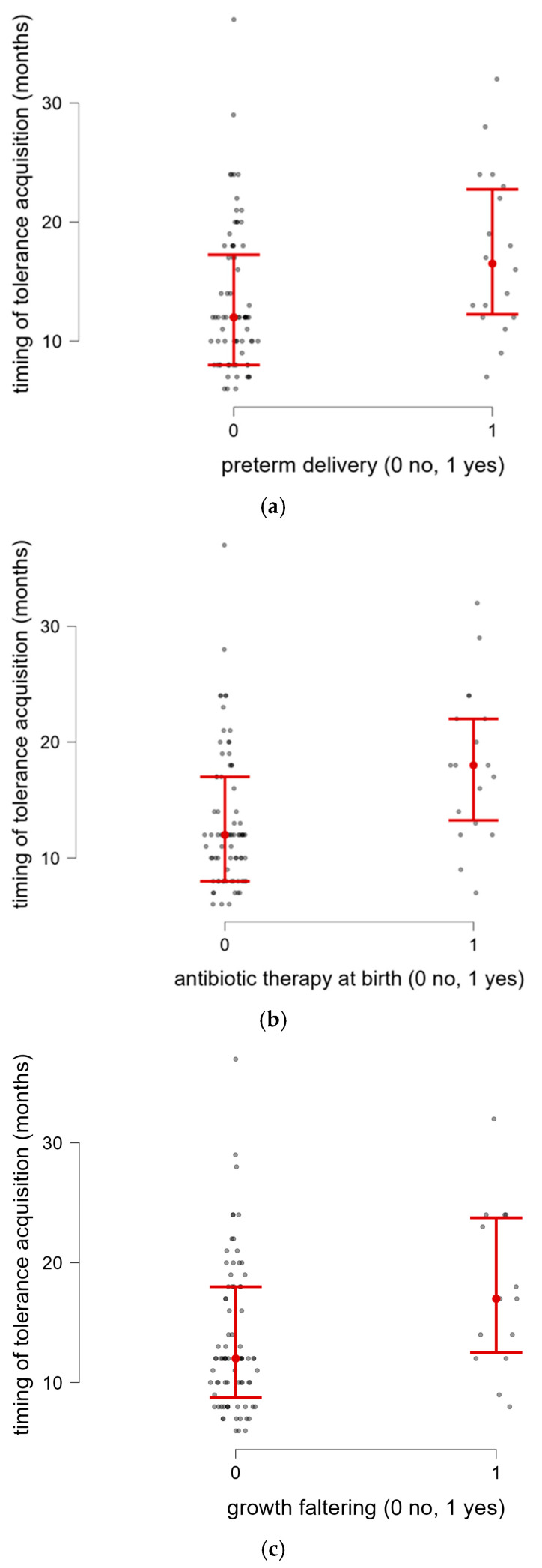
(**a**) Timing of tolerance acquisition in relation to preterm delivery. (**b**) Timing of tolerance acquisition in relation to antibiotic treatment within the first seven days of age. (**c**) Timing of tolerance acquisition in relation to growth faltering.

**Figure 4 nutrients-18-00095-f004:**
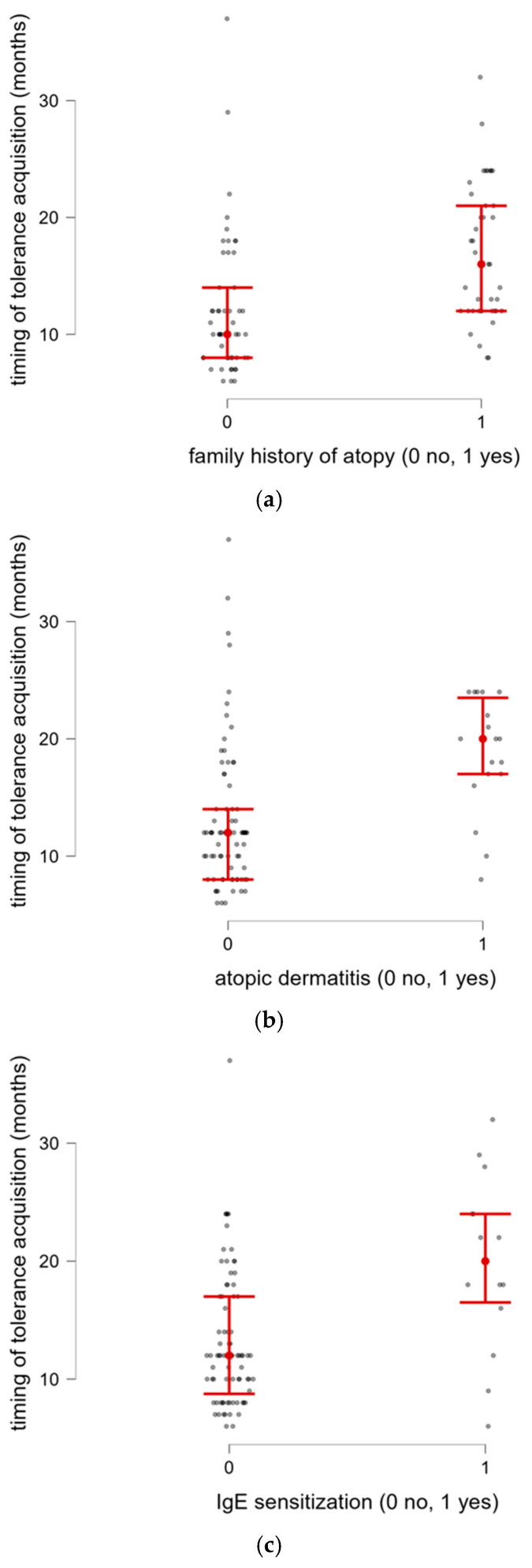
(**a**) Timing of tolerance acquisition in relation to family history of allergy. (**b**) Timing of tolerance acquisition in relation to patient’s atopic dermatitis. (**c**) Timing of tolerance acquisition in relation to results of allergy tests.

**Figure 5 nutrients-18-00095-f005:**
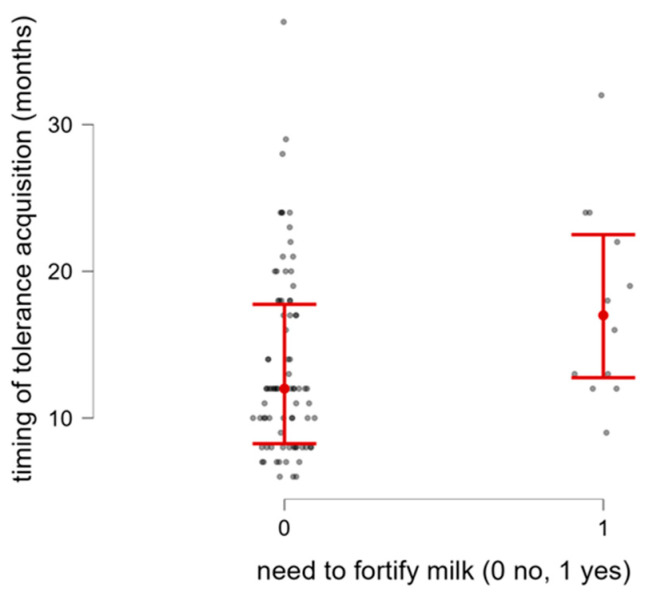
Timing of tolerance acquisition in relation to fortification of milk in the first weeks of life.

**Figure 6 nutrients-18-00095-f006:**
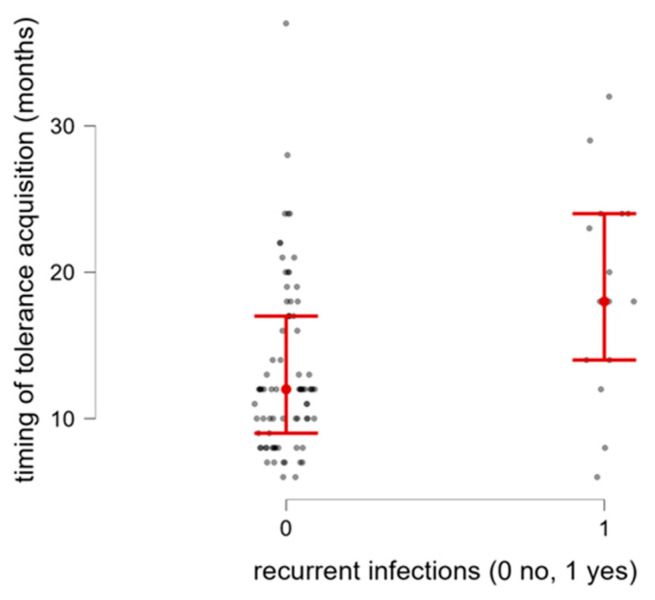
Timing of tolerance acquisition in relation to occurrence of recurrent infections.

**Figure 7 nutrients-18-00095-f007:**
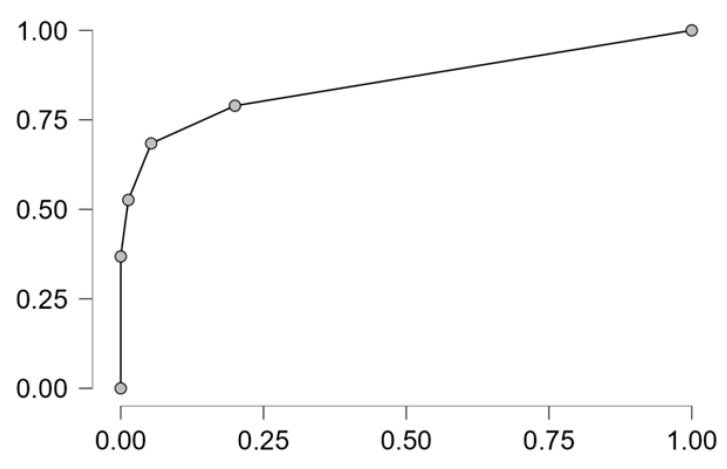
ROC curve of logistic regression model.

**Table 1 nutrients-18-00095-t001:** Demographic and clinical findings of the study population.

		Total (*n*)
**Age of onset [months] (median and IQR)**	2.9 (1.9–4.7)	94
**Male sex**	57%	54
**Preterm**	19%	18
◦ extremely-very preterm	5%	5
◦ moderate-late preterm	14%	13
**Growth faltering**	15%	14
**Antibiotic treatment within the first seven days of age**	19%	18
**Fortification of milk in the first weeks of life**	13%	12
**Family history of atopy**	44%	41
**Atopic dermatitis (AD)**	19%	18
**Culprit food**		
◦ cow milk (CM)	100%	94
◦ others (e.g., egg, soy, wheat)	-	0
◦ multiple food	-	0
**Clinical features**		
◦ rectal bleeding	100%	94
◦ recurrent vomit	6%	6
◦ crying/colic	5%	5
◦ diarrhea	3%	3
**Type of feeding**		
◦ breastfeeding	62%	58
◦ formula feeding	18%	17
◦ breast and formula feeding	20%	19
**Type of elimination diet**		
◦ maternal CM free diet	62%	58
◦ extensive CM based hydrolyzed formula	32%	30
◦ amino acid formula [AAF]	5%	5
◦ rice-based HF	1%	1
**IgE sensitization**	15%	14
◦ positive SPTs	5%	5
◦ positive specific IgE	10%	9
**Recurrent infections**	16%	15
**Rotavirus vaccination**	54%	51

**Table 2 nutrients-18-00095-t002:** Demographic and clinical differences between the Varese and Milan cohorts.

	Varese Cohort (*n* = 61)	Milan (*n* = 33)
**Age of onset [months] (median and IQR)**	3.2 (2–5)	2.4 (1.3–4.2)
**Male sex**	30 (49%)	24 (73%)
**Preterm**	7 (11%)	11 (33%)
**Growth faltering**	10 (16%)	4 (12%)
**Antibiotic treatment within the first seven days of age**	10 (16%)	8 (24%)
**Fortification of milk in the first weeks of life**	5 (8%)	7 (21%)
**Family history of atopy**	24 (39%)	17 (52%)
**Atopic dermatitis (AD)**	14 (23%)	4 (12%)
**Type of feeding**		
◦ breastfeeding	48 (79%)	10 (30%)
◦ formula feeding	6 (10%)	11 (33%)
◦ breast and formula feeding	7 (11%)	12 (37%)
**Type of elimination diet**		
◦ maternal CM free diet	48 (79%)	10 (30%)
◦ extensive CM based hydrolyzed formula	12 (20%)	18 (55%)
◦ amino acid formula [AAF]	0	5 (15%)
◦ rice-based HF	1 (1%)	0
**IgE sensitization**	10 (16%)	4 (12%)
◦ positive SPTs	8 (13%)	1 (3%)
◦ positive specific IgE	2 (3%)	3 (9%)
**Recurrent infections**	10 (16%)	5 (15%)
**Rotavirus vaccination**	35 (57%)	16 (48%)

**Table 3 nutrients-18-00095-t003:** Food tolerance.

		Total (*n*)
**Age of tolerance acquistion [months] (median and IQR)**	12 (9.7–18)	94
**Food reintroduction [cow’s milk (CM)]**		
◦ ladder modality	84%	79
◦ CM	11%	10
◦ dairy products	4%	4
◦ reintroduction of CM in maternal diet	1%	1

**Table 4 nutrients-18-00095-t004:** Factors that may potentially function as predictors of late tolerance acquisition to cow’s milk.

	Odds Ratio (OR)	Odds Ratio (OR) 95%CI Lower Bound	Odds Ratio (OR) 95%CI Upper Bound	*p*-Value
**Family history of atopy**	**5.433**	**1.164**	**25.356**	**0.031**
**Atopic dermatitis**	**8.211**	**1.659**	**40.653**	**0.010**
Preterm delivery	4.754	0.548	41.273	0.157
Fortification of milk in the first weeks of life	0.138	0.011	1.674	0.120
Growth faltering	1.528	0.264	8.824	0.636
**Rectal bleeding duration >18 days before starting elimination diet**	**5.948**	**1.276**	**27.720**	**0.023**
Antibiotic treatment within the first seven days of age	1.437	0.201	10.266	0.718
Recurrent infections	1.654	0.307	8.921	0.233
**IgE sensitization**	**6.352**	**1.151**	**35.043**	**0.034**

The bold formatting is used within the table itself to highlight statistically significant variables.

**Table 5 nutrients-18-00095-t005:** Performance metrics of logistic regression model.

accuracy	0.883
AUC	0.888
sensitivity	0.526
specificity	0.973
precision	0.833

## Data Availability

The data presented in this study are available on request from the corresponding author in order to guarantee the privacy of the participants, whose research data are confidential.

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
