# Peer review of "Identifying Predictors for the Acquisition of Tolerance to Cow’s Milk Protein in Infants with Food Protein-Induced Allergic Proctocolitis (FPIAP): Multifactorial Analysis of Two Italian Cohorts"

_nutrients, 2025, doi:10.3390/nu18010095_

Round 1

Reviewer 1 Report

Comments and Suggestions for Authors

Thank you for the invitation to review the manuscript titled “ Predictors for the acquisition of tolerance to cow’s milk protein 2 in infants with food protein-induced allergic proctocolitis 3 (FPIAP): multifactorial analysis of two Italian cohorts” which was submitted to Nutrients as an article. Therein, the authors aimed to “ analyse clinical features and predictive factors for the acquisition of tolerance to cow’s milk (CM) in infants with FPIAP” (copied verbatim from the manuscript) in a cross-sectional study of 94 infants.

Major comments

  1. There are many figures, not all of which are strictly necessary.
  2. Subgroup analyses are often based on very small numbers. The authors are encouraged to critically consider the impact of such analyses on the interpretation, and the rigour of their analyses.
  3. To my mind, the most interesting analysis is that which is presented in Section 3.12: Predictors of late acquisition of CM tolerance. Focusing the manuscript on this section, and thus removing many of the subgroup analyses, would result in a more succinct and concise manuscript with a practical take away message for the reader.

Minor comments

  1. Abstract, Methods & Results: The mention of milk fortification is confusing, as fortification in the area of nutrition commonly refers to adding micronutrients to a food product, such as iodine fortification of flour. However, I don’t believe that this is what the authors meant. Please revise to a different descriptor.
  2. Abstract, Methods: Be specific about the type of bleeding.
  3. Abstract, Methods: Why was rotavirus vaccination analysed?
  4. Abstract: There is no mention of the age of the infants. Similarly, how pre-term is pre-term?
  5. Abstract: If the median age at tolerance was age 12 months, it must be mentioned at what age the infants were diagnosed, from which the interval between diagnosis and tolerance can be derived.
  6. Abstract, Results: The definition of late tolerance belongs in the Methods. The number of infants who were in this group needs to be reported.
  7. Abstract, Results: Add the 95%CI for the OR. It is insufficient to simply report p-values, particularly with a relatively small sample size. The same comment applies to the Results section in the main text.
  8. Methods, Line 128: The term “failure to thrive” is falling out of favour. Please revise to “growth faltering.”
  9. Results, Line 153-154: Please provide some description about the demographic differences between the infants from Varese and Milan. Similarly, were practices standardized between the two sites?
  10. Results, Line 160, and Figure 3: Here, the authors refer to biological sex, not gender. Infants and children under age 3 years have not yet formed a gender identity.
  11. Results, Line 163: Revise the abbreviation for atopic dermatitis from DA to AD.
  12. Figure 2a: The is no x-axis title.
  13. Results, Lines 207-8: While there may indeed be no association (not impact; this analysis is not possible in a cross-sectional study) between SGA and CM tolerance, the authors have not considered that this null finding is possibly (and indeed, probably) due to the fact that there were very few cases (n=6) of SGA.
  14. Tables: Minor formatting issues, such as lack of alignment of columns.
  15. Discussion, Lines 324-5: As the authors presented a cross-sectional study, it is not correct to suggest that they “investigated predictive factors of tolerance”
  16. The manuscript contains many grammatical and linguistic errors. While none of these errors impede comprehension, they are nonetheless distracting and must be corrected. Similarly, the Methods section alternates between past and present tense. Given that this study is completed, this section should be written in the past tense.

Author Response

Response to Reviewer1

We sincerely thank the Reviewer for the thorough and constructive evaluation of our manuscript. All comments have been addressed, and the corresponding revisions are detailed in the point-by-point responses below. 

2. Questions for General Evaluation

Reviewer’s Evaluation

Response and Revisions

Does the introduction provide sufficient background and include all relevant references?

Can be improved

We have revised the Introduction to improve clarity and strengthen the background, adding relevant references where appropriate.

Is the research design appropriate?

Can be improved

We acknowledge the reviewer’s indication that the research design could be refined. We have clarified the rationale of the cross-sectional design and adjusted the wording throughout the manuscript to better reflect its descriptive nature.

Are the methods adequately described?

Must be improved

We appreciate this comment and have substantially revised the Methods section to improve clarity and consistency of terminology.

Are the results clearly presented?

Must be improved

We thank the Reviewer for this observation. The Results section has been reorganized and clarified, with improved descriptions, consistent terminology, and removal of analyses not adequately powered.

Are the conclusions supported by the results?

Must be improved

We agree with the reviewer’s assessment. The Conclusions have been revised to ensure that they are supported by the results and to clarify the descriptive and exploratory scope of the findings.

Are all figures and tables clear and well-presented?

Can be improved

We thank the Reviewer for this feedback. Figures and tables have been revised for clarity, alignment, and overall presentation quality.

3. Point-by-point response to Comments and Suggestions for Authors

Major comments

Comments 1: There are many figures, not all of which are strictly necessary.

Response 1: We thank the Reviewer for this helpful observation. In the revised manuscript, we have critically reviewed all figures and removed those that were redundant or added limited value to the main message. Specifically, Figures 2a, 3, 7a, 7b, 10, 11 have been removed in order to ensure greater readability. We believe this change leads to a more focused presentation of our findings.

Comments 2: Subgroup analyses are often based on very small numbers. The authors are encouraged to critically consider the impact of such analyses on the interpretation, and the rigour of their analyses.

Response 2: We appreciate the Reviewer’s concern regarding the limited sample size in some subgroup analyses. We have carefully revised the manuscript to address this issue. All subgroup analyses now report sample sizes, and their limitations are acknowledged in the Discussion section.

In addition, we have removed SGA subgroup analysis due to the particularly small sample size that did not contribute meaningfully to the interpretation of the results. We agree that this improves the rigour of our study.

Comments 3: To my mind, the most interesting analysis is that which is presented in Section 3.12: Predictors of late acquisition of CM tolerance. Focusing the manuscript on this section, and thus removing many of the subgroup analyses, would result in a more succinct and concise manuscript with a practical take away message for the reader.

Response 3: We thank the Reviewer for highlighting the relevance of Section 3.12 (corresponding to section 3.10 in the revised manuscript). As suggested, we have expanded this section by providing a comprehensive description of the main demographic and clinical features of infants exhibiting late CM tolerance. This addition was intended to ensure a more accurate characterization of this subgroup and to support a more critical and complete interpretation of our findings. At the same time, we have structurally revised the sections concerning the subgroup association analyses by consolidating them into broader, thematically coherent main sections. Specifically, neonatal features were integrated with growth data (including growth faltering), whereas atopic features were grouped together with IgE sensitization. In addition, the subgroup analyses involving SGA has been removed, as explained and justified in the previous response (response 2). This change was undertaken to enhance the clarity, readability, and overall flow of the text and to provide a clearer take-home message for readers.

Minor comments

Comments 1: The mention of milk fortification is confusing, as fortification in the area of nutrition commonly refers to adding micronutrients to a food product, such as iodine fortification of flour. However, I don’t believe that this is what the authors meant. Please revise to a different descriptor.

Response 1: We appreciate the Reviewer for pointing out the ambiguity regarding the term milk fortification. We agree that the wording used in the original manuscript may have generated confusion. In our study, milk fortification refers to the deliberate addition of essential micronutrients—namely vitamins and minerals—to human milk, in accordance with the widely accepted definition of food fortification provided by the World Health Organization (WHO).

To avoid ambiguity, we have modified the terminology and we explicitly refer to this process as micronutrient fortification of human milk. We have also added a clarifying statement in the Methods section to guarantee consistency with the WHO definition.

Comments 2: Be specific about the type of bleeding

Response 2: We thank the Reviewer for this comment. In our study, bleeding specifically referred to rectal bleeding, defined as the presence of visible traces of blood in the stool, even in minimal amounts. This included both stools containing small streaks of blood and frankly blood-stained stools. We have added this definition clearly in both the Method section and in the Abstract (in a short form) of the revised manuscript. We have also explicitly specified the rectal nature of the bleeding in all parts of the manuscript where the term ‘bleeding’ was previously used without this clarification.

Comments 3: Why was rotavirus vaccination analysed?

Response 3: We thank the Reviewer for this question. We decided to include Rotavirus vaccination as a variable because some parents reported that rectal bleeding appeared within a few days after the administration of the Rotavirus vaccine. Based on these observations, we aimed to explore whether a possible correlation could exist, even indirectly, considering that a probable immunological or inflammatory mechanism may theoretically explain this association in selected cases.

Although our study was not designed to evaluate pathophysiological correlations, we considered it important to assess this association descriptively. This clarification has been incorporated into the Methods section. Should any signal have emerged, it would have required confirmation in future studies with the aim of verifying underlying immune or inflammatory pathways or other mechanisms.

Comments 4: There is no mention of the age of the infants. Similarly, how pre-term is pre-term?

Response 4: We thank the Reviewer for pointing out this missing data. We have added the median age with IQR of the infants in the Abstract. We have also specified our definition of pre-term infants (< 37 weeks of gestational age), consistent with the criteria used in the manuscript.

Comments 5: If the median age at tolerance was age 12 months, it must be mentioned at what age the infants were diagnosed, from which the interval between diagnosis and tolerance can be derived.

Response 5: We thank the Reviewer for this helpful suggestion. We have reported the median age with IQR at diagnosis in the Abstract, allowing the reader to infer the interval between diagnosis and CM tolerance acquisition.

Comments 6: The definition of late tolerance belongs in the Methods. The number of infants who were in this group needs to be reported.

Response 6: The definition of late tolerance has been moved to the Methods, where it is more appropriate, and this correction has been also issued in the Abstract. In the Results, we have reported the number of infants classified in this group.

Comments 7: Add the 95%CI for the OR. It is insufficient to simply report p-values, particularly with a relatively small sample size. The same comment applies to the Results section in the main text.

Response 7: We thank the Reviewer for raising this important point. We have added 95% confidence intervals for all ORs reported in the Abstract and in the Results section. This provides a statistically more appropriate assessment of our data.

Comments 8: The term “failure to thrive” is falling out of favour. Please revise to “growth faltering.”

Response 8: We thank the Reviewer for this suggestion. The term has been replaced with “growth faltering”, according to the current terminology.

Comments 9: Please provide some description about the demographic differences between the infants from Varese and Milan. Similarly, were practices standardized between the two sites?

Response 9: We thank the Reviewer for this request. We have added a description of demographic and clinical differences between the two cohorts (Varese and Milan). As a preliminary step, clinical and diagnostic procedures were harmonized across the two centres to guarantee methodological consistency and minimize procedural biases with the only difference concerning the distribution of allergy testing modalities. Indeed, skin prick tests were more commonly performed in the Varese cohort, whereas specific IgE measurement was more frequently used in the Milan cohort, possibly related to the different outpatient clinics in the two centers (Pediatric Gastroenterology clinic in Varese and Pediatric Allergology clinic in Milan). We have also added a clarifying statement in the Methods section.

Comments 10: Here, the authors refer to biological sex, not gender. Infants and children under age 3 years have not yet formed a gender identity.

Response 10: We thank the Reviewer for this clarification. In all sections, we now refer exclusively to biological sex, considering that gender identity is not applicable to infants and children under age 3 years.

Comments 11: Revise abbreviation for atopic dermatitis from DA to AD.

Response 11: Thank you for noticing this. The abbreviation has been corrected in the manuscript.

Comments 12: Figure 2a: “There is no x-axis title.”

Response 12: The Figure 2a has been removed in the revised version in order to ensure greater clarity and readability, as reported in response to Major comment 1.

Comments 13: While there may indeed be no association (not impact; this analysis is not possible in a cross-sectional study) between SGA and CM tolerance, the authors have not considered that this null finding is possibly (and indeed, probably) due to the fact that there were very few cases (n=6) of SGA.

Response 13: We thank the Reviewer for this insightful observation. We fully agree that, given the very limited number of SGA infants in our cohort, any analysis involving this subgroup would lack sufficient statistical power, without allowing us to yield methodologically robust conclusions. Although we initially included this analysis because we believed it could provide interesting insights into the potential role of SGA, we acknowledge that, due to the small sample size of our cohort, its statistical solidity is insufficient.

For this reason, we have removed the subsection concerning SGA from the Results and have adjusted the corresponding parts of the manuscript accordingly. We believe this revision improves the methodological rigor of the paper.

Comments 14: Tables: Minor formatting issues, such as lack of alignment.

Response 14: We thank the Reviewer for bringing this to our attention. We have revised the formatting of all tables to improve alignment and readability.

Comments 15: As the authors presented a cross-sectional study, it is not correct to suggest that they “investigated predictive factors of tolerance”

Response 15: We thank the Reviewer for this important comment. We agree that, within the Discussion, the expression “investigated predictive factors” could suggest a temporal or causal inference, which is not appropriate for a cross-sectional study. For this reason, we have corrected it as follows “explored factors associated with tolerance that may potentially function as predictors”.

Regarding the use of the term “predictive factors” in the title and other sections, our intention was not to imply causality or formal predictive modelling, but rather to describe the conceptual aim of the study—namely, to explore characteristics that could potentially function as predictors of tolerance acquisition in future longitudinal research. For clarity, we have now explicitly added this explanation in Discussion sections and we have slightly modified the title. These additions reiterate that the study is descriptive and that any potential predictive role of the identified factors would require confirmation in prospective studies.
In the Conclusions, we have also specified that the role of the identified factors should be interpreted as potentially predictive, through the use of the term Potential Predictors.

Comments 16: The manuscript contains many grammatical and linguistic errors. While none of these errors impede comprehension, they are nonetheless distracting and must be corrected. Similarly, the Methods section alternates between past and present tense. Given that this study is completed, this section should be written in the past tense.

Response 16: We thank the Reviewer for this comment. We have revised the entire manuscript to correct grammatical and linguistic errors. In addition, we have rewritten the Methods section consistently in the past tense, considering that the study has been completed. We are confident that these revisions improve the readability of the manuscript.

4. Response to Comments on the Quality of English Language

We thank the Reviewer for highlighting the linguistic and grammatical issues. We have revised the manuscript to improve the quality of the English Language, correcting grammatical and stylistic inaccuracies and adopting clearer phrasing.

5. Additional clarifications

We thank the Reviewer once again for the careful and accurate evaluation, which substantially improved the Methods and Results sections, statistical accuracy, linguistic adequacy and clarity of our manuscript.

Reviewer 2 Report

Comments and Suggestions for Authors

This is a study focused on the risk factors for allergic proctocolitis and most interesting on its duration. One of them was found to be the bleeding duration before starting diet. The rest of them were interconnected, all related to atopy (family history of atopy, concomitant atopic dermatitis, bleeding duration and IgE sensitization).

I would suggest to extent the part of the introduction on the elimination diet, since it is not clear at this point that maternal diet is included in this practice (it is reported much later in the results).

I would also like to have a clear explanation on the examination of Rotavirus vaccination as a risk factor for FPIAP. It is not reported clear why this specific vaccine was taken under consideration.

I think that replying to these minor notes can improve the article. I would also avoid the tiny paragraphs in the introduction and omit the graphics that present no significant difference.

Author Response

1. Response to Reviewer 

We thank the Reviewer for the careful evaluation of our manuscript and the constructive comments, which have helped us improve the overall quality of the work. Below we provide a point-by-point response.

2. Questions for General Evaluation

Reviewer’s Evaluation

Response and Revisions

Does the introduction provide sufficient background and include all relevant references?

Yes

Thanks

Is the research design appropriate?

Yes

Thanks

Are the methods adequately described?

Yes

Thanks

Are the results clearly presented?

Yes

Thanks

Are the conclusions supported by the results?

Yes

Thanks

Are all figures and tables clear and well-presented?

Yes

Thanks

3. Point-by-point response to Comments and Suggestions for Authors

Comments 1: I would suggest to extent the part of the introduction on the elimination diet, since it is not clear at this point that maternal diet is included in this practice (it is reported much later in the results)

Response 1: We thank the Reviewer for this relevant observation. We have now expanded the section of the Introduction dedicated to the elimination diet, specifying from the outset that the dietary intervention includes maternal elimination diet in breastfed infants. This clarification ensures consistency with the details later presented in the Results.

Comments 2: I would also like to have a clear explanation on the examination of Rotavirus vaccination as a risk factor for FPIAP. It is not reported clear why this specific vaccine was taken under consideration.

Response 2: We thank the reviewer for this question. We decided to include Rotavirus vaccination as a variable because some parents reported that rectal bleeding appeared within a few days after the administration of the rotavirus vaccine. Based on these observations, we aimed to explore whether a possible correlation could exist, even indirectly, considering that a probable immunological or inflammatory mechanism may theoretically explain this association. Although our study was not designed to evaluate pathophysiological correlations, we considered it important to assess this association descriptively. This clarification has been incorporated into the Methods section. Should any signal have emerged, it would have required confirmation in future studies with the aim of verifying underlying immune or inflammatory pathways or other mechanism.

Comments 3: I would also avoid the tiny paragraphs in the introduction and omit the graphics that present no significant difference.

Response 3: We thank the Reviewer for these stylistic suggestions. The short paragraphs in the Introduction have been merged to guarantee a more fluent structure. The figures showing non-significant differences or non-relevant data (2a, 3, 7a, 7b, 10, 11) have been removed, as suggested, to maintain focus on the most statistically significant results

We thank the Reviewer once again for the constructive insights, which have substantially improved the clarity and coherence of our manuscript.

Reviewer 3 Report

Comments and Suggestions for Authors

This is an interesting observational study looking at factors that may relate to differential acquisition of tolerance to non-IgE mediated CM allergy related to intestinal immune mechanisms. I think the authors might give some suggestions that could help parents and doctors of patients with bloody diarrhea probably caused by immune factors of allergy, consider what and how to measure the patient's progress and likely tolerance acquisition. And consider possible causes of intolerance. 

Author Response

1. Response to Reviewer 

We thank the Reviewer for the positive overall assessment of our observational study and for the constructive suggestions.

2. Questions for General Evaluation

Reviewer’s Evaluation

Response and Revisions

Does the introduction provide sufficient background and include all relevant references?

Yes

Thanks

Is the research design appropriate?

Yes

Thanks

Are the methods adequately described?

Can be improved

The Methods section has been refined to improve overall readability

Are the results clearly presented?

Can be improved

The Results section has been revised to present the main findings in a clearer way, highlighting the factors most relevant to clinical interpretation

Are the conclusions supported by the results?

Yes

Thanks

Are all figures and tables clear and well-presented?

Yes

Thanks

3. Point-by-point response to Comments and Suggestions for Authors

Comments 1: I think the authors might give some suggestions that could help parents and doctors of patients with bloody diarrhea probably caused by immune factors of allergy, consider what and how to measure the patient's progress and likely tolerance acquisition. And consider possible causes of intolerance.

Response 1: According to Reviewer’s recommendation, we have expanded the Discussion to include practical considerations that may support clinicians and parents in in the clinical approach and follow-up of infants with FPIAP. Specifically, we have incorporated a concise overview of relevant factors that should be monitored over time, as they may potentially influence the timing of CM tolerance acquisition (e.g., bleeding duration before elimination diet, recurrent infections, atopic manifestations, growth parameters).

Once again, we sincerely thank the Reviewer for thoughtful comments that have helped us to translate our findings into useful tools for real-world clinical management of FPIAP.

Round 2

Reviewer 1 Report

Comments and Suggestions for Authors

Thank you for the invitation to re-review this manuscript. While much improved, some issues remain.

Section 2.2 Data collection

Despite the name of this section, the text does not describe data collection. Rather, this text details the operationalisation of various variables used by the authors.

Figure 1

This figure is difficult to read at 100% magnification.

Table 2.

This table tells us that there were notable differences between the two cohorts: sex distribution, prematurity, antibiotic treatment in the first 7 days of life, micronutrient fortification, breastfeeding and more. Yet the authors did not account for these differences. Please repeat the analyses performed, but for each group independently. Numbers will likely be small, and 95%CI wide. But, such analyses will also provide insight into the directionality of the odds ratios.

Table 4.

Many of the 95%CI are very wide. While the results are statistically significant, they are not terribly meaningful.

Author Response

Response to Reviewer

We sincerely thank the reviewer for the constructive comments, which have allowed us to improve the clarity and rigor of the manuscript. We address each point below.

2. Questions for General Evaluation

  Reviewer’s Evaluation

Response and Revisions

Does the introduction provide sufficient background and include all relevant references?

 Can be improved

We have revised the Introduction to improve clarity and strengthen the background, adding relevant references where appropriate.

Is the research design appropriate?

 Can be improved

We acknowledge the Reviewer’s indication that the research design could be refined. We have clarified the rationale of the cross-sectional design and adjusted the wording throughout the manuscript to better reflect its descriptive nature.

Are the methods adequately described?

  Must be improved

We appreciate this comment and have substantially revised the Methods section to improve clarity and consistency of terminology.

Are the results clearly presented?

 Must be improved

We thank the Reviewer for this observation. The Results section has been reorganized and clarified, with improved descriptions, consistent terminology, and removal of analyses not adequately powered.

Are the conclusions supported by the results?

   Must be improved

We agree with the Reviewer’s assessment. The Conclusions have been revised to ensure that they are supported by the results and to clarify the descriptive and exploratory scope of the findings.

Are all figures and tables clear and well-presented?

  Can be improved

We thank the Reviewer for this feedback. Figures and tables have been revised for clarity, alignment, and overall presentation quality.

3. Point-by-point response to Comments and Suggestions for Authors

Comments 1: Section 2.2 Data collection.

Despite the name of this section, the text does not describe data collection. Rather, this text details the operationalisation of various variables used by the authors..

Response 1: We thank the Reviewer for this observation. To better reflect the content of the section, the title has been revised from “Data Collection” to “Variable Specification and Measurement”. This section now clearly describes the operationalisation of all variables included in the analyses, providing readers with a precise understanding of how each variable was defined and measured.

Comments 2: Figure 1 This figure is difficult to read at 100% magnification.

Response 2: We appreciate the Reviewer’s comment regarding Figure 1. The figure has been reformatted to improve readability at 100% magnification while preserving the original layout and content. These changes ensure that the figure is fully accessible and interpretable without altering the presented results.

Comments 3: Table 2.

This table tells us that there were notable differences between the two cohorts: sex distribution, prematurity, antibiotic treatment in the first 7 days of life, micronutrient fortification, breastfeeding and more. Yet the authors did not account for these differences. Please repeat the analyses performed, but for each group independently. Numbers will likely be small, and 95%CI wide. But, such analyses will also provide insight into the directionality of the odds ratios.

Response 3:  We thank the Reviewer for highlighting the differences between the two cohorts. In order to evaluate the directionality of effects, as suggested by the Reviewer, cohort-specific logistic regression analyses were performed independently for each group. Due to the small number of events per cohort and small sample sizes, the resulting estimates were highly unstable, with wide confidence intervals. While the absolute values of the odds ratios are not interpretable, the analyses provide directional assessment of each association. All cohort-specific analyses are cited in the Results section and provided in detail in the Supplementary Tables, ensuring full transparency and allowing readers to assess the consistency of associations across cohorts. This limitation has been explicitly discussed in the Discussion section, ensuring that readers are aware of its implications for interpreting the findings.

Comments 4: Table 4.

Many of the 95%CI are very wide. While the results are statistically significant, they are not terribly meaningful.

Response 4: We appreciate the Reviewer’s observation regarding the width of confidence intervals in Table 4. As noted, many 95% CIs are wide due to limited sample sizes and a low number of events. While some associations reached statistical significance, we have now explicitly stated in the Discussion that these results should be interpreted with caution. By clarifying this limitation, we aim to provide an accurate and transparent interpretation of the findings, in line with the reviewer’s recommendation.

We sincerely appreciate the Reviewer’s thoughtful and detailed comments, which have been extremely helpful in enhancing the rigor, transparency, and presentation of our work